# Vocational Domain Identification with Machine Learning and Natural Language Processing on Wikipedia Text: Error Analysis and Class Balancing †

Maria Nefeli Nikiforos *, Konstantina Deliveri, Katia Lida Kermanidis and Adamantia Pateli

Department of Informatics, Ionian University, 49132 Corfu, Greece; p15deli@ionio.gr (K.D.);
kerman@ionio.gr (K.L.K.); pateli@ionio.gr (A.P.)
* Correspondence: nefeli.nikiforos@ionio.gr
† This paper is an extended version of our paper published in 2022 17th International Workshop on Semantic and Social Media Adaptation & Personalization (SMAP), Online Event, 3–4 November 2022.

**Abstract:** Highly-skilled migrants and refugees finding employment in low-skill vocations, despite professional qualifications and educational backgrounds, has become a global tendency, mainly due to the language barrier. Employment prospects for displaced communities are mostly decided by their knowledge of the sublanguage of the vocational domain they are interested in working. Common vocational domains include agriculture, cooking, crafting, construction, and hospitality. The increasing amount of user-generated content in wikis and social networks provides a valuable source of data for data mining, natural language processing, and machine learning applications. This paper extends the contribution of the authors' previous research on automatic vocational domain identification by further analyzing the results of machine learning experiments with a domain-specific textual data set while considering two research directions: a. prediction analysis and b. data balancing. Wrong prediction analysis and the features that contributed to misclassification, along with correct prediction analysis and the features that were the most dominant, contributed to the identification of a primary set of terms for the vocational domains. Data balancing techniques were applied on the data set to observe their impact on the performance of the classification model. A novel four-step methodology was proposed in this paper for the first time, which consists of successive applications of SMOTE oversampling on imbalanced data. Data oversampling obtained better results than data undersampling in imbalanced data sets, while hybrid approaches performed reasonably well.

**Keywords:** natural language processing; social text mining; machine learning; vocational domain identification; vocational language; error analysis; class balancing

## 1. Introduction

### 1.1. Vocational Domains for Migrants and Refugees

Migrant employees face multiple challenges deriving from discrimination due to their country of origin, nationality, culture, sex, etc. [1–4]. For women in particular, finding employment in high-skill vocations, besides teaching and nursing, has been observed to be especially difficult [1,2,5]. A deciding factor regarding the prospects of employment for displaced communities, such as migrants and refugees, is the knowledge of not the language of their host country in general, but specifically of the sublanguage of the vocational domain they are interested in working. As a result, highly-skilled migrants and refugees finding employment in low-skill vocations, despite their professional qualifications and educational backgrounds, has become a global tendency, with the language barrier being one of the most important factors [1–4,6].

The scope of vocational domains for displaced communities and analyses on their situations in the host country and in their country of origin were examined in the recent literature, which considered the impact on their work-life balance [2–4]. Both high-skill and

low-skill vocations in hospitality, cleaning, manufacturing, retail, crafting, and agriculture were the most common vocational domains in which migrants and refugees sought and found employment, according to the findings of the recent research [1–3,6–8]. It is also important to note that unemployment usually affects the displaced communities more than the natives [9]. Overworking, however, due to low-paid jobs, thrives, as migrants and refugees struggle to increase their earnings in their efforts to maintain living standards, afford childcare, and be able to send remittances to remaining family in their country of origin [2,7].

### 1.2. Wikipedia and Social Networks

Due to the expansion of the user base of wikis and social networks in the last decade, user-generated content has increased in great amounts. This content provides a valuable source of data for various tasks and applications in data mining, natural language processing (NLP), and machine learning. Wikipedia (https://en.wikipedia.org/wiki/Main_Page, accessed on 20 May 2023) is an open data wiki that covers a wide scope of topic-related articles written in many languages [10]. Wikipedia's content generation is a constant collective process derived from the collaboration of its users [11]; as of April 2023, there were approximately 6.6 million Wikipedia articles written in English.

### 1.3. Class Imbalance Problem

Imbalanced data sets, in regard to class distribution among their examples, present several challenges in data mining and machine learning tasks. More specifically, the number of examples representing the class of interest is considerably smaller than the ones of the other classes. As a result, standard classification algorithms have a bias towards the majority class, and, consequently, they tend to misclassify the minority class examples [12]. Most commonly, the class imbalance problem is related to binary classification, although it is not uncommon for it to emerge in multi-class problems (such as in this paper); since there are more than one minority class, it is more challenging to solve. The class imbalance problem is an issue that affects various aspects of real-world applications that are based on classification due to the fact that the minority class examples are the most difficult to obtain from real data, especially from user-generated content from wikis and social networks, which has led a large community of researchers to examine ways to address it [12–18].

### 1.4. Contributions

This paper extends the contribution of the authors' previous research [19] by exploring the various potential directions deriving from it. The results of the machine learning experiments with a domain-specific textual data set that was created and preprocessed as described in [19] were further processed and analyzed with the consideration of two research directions: a. prediction analysis and b. data balancing.

More specifically, regarding the prediction analysis, important conclusions were drawn from examining which examples were classified wrongly for each class (wrong predictions) by the *Gradient Boosted Trees* model, which managed to classify most of the examples correctly, as well as which distinct features contributed to their misclassification. In the same line of thought, regarding the correctly classified examples (correct predictions), the examination of the features that were the most dominant and led to the correct classifications for each class contributed to the identification of a primary set of terms highlighting the terminology of the vocational domains.

Regarding the data balancing, oversampling, and undersampling techniques, both separate techniques and combined techniques, as a hybrid approach, were applied on the data set in order to observe their impact (positive or negative) on the performance of the *Random Forest* and *AdaBoost* model. A novel and original four-step methodology was proposed in this paper and used for data balancing for the first time, to the best of the authors' knowledge. It consists of successive applications of SMOTE oversampling on imbalanced data in order to balance them by considering which class is the minority class in each iteration. By running the experiments while following this methodology, the

impact of every class distribution, from completely imbalanced to completely balanced data, on the performance of the machine learning model could be examined thoroughly. This process of data balancing enabled the comparison of the performance of this model with balanced data to the performance of the same model with imbalanced data from the previous research [19]. The findings derived from the machine learning experiments of this paper are in accordance with those of the relevant literature [12,17] in terms that data oversampling obtaining better results than data undersampling in imbalanced data sets, while the hybrid approaches performed reasonably well.

### 1.5. Structure

The structure of this paper is as follows. Section 2 presents past related work on a. domain identification on textual data, b. data scraping from social text, and c. data oversampling, undersampling, and hybrid approaches. Section 3 describes the stages of data set creation and preprocessing, as well as the feature extraction process. Section 4 presents the research direction of the prediction analysis, including both wrong and correct predictions of the *Gradient Boosted Trees* model. Section 5 presents the research direction of the data balancing, including the novel four-step methodology for successive SMOTE oversampling, as well as experiments with data undersampling and a hybrid approach. Section 6 concludes the paper, discusses the most important findings, and draws directions for future work.

## 2. Related Work

In this Section, the recent literature on domain identification on textual data, including news articles, technical text, open data, and Wikipedia articles, is presented. Research on data scraping from social text, sourced from social networks and Wikipedia, is also described. Finally, the findings of related work regarding data oversampling, undersampling, and hybrid approaches are also analyzed.

### 2.1. Domain Identification on Textual Data

Domain identification performed on textual data, including news articles, social media posts, and social text data sets in general, remains an open problem and a very challenging task for researchers. The vast domain diversity, along with the particular sublanguage and terminology, present several challenges when undertaking domain identification on textual and linguistic data.

#### 2.1.1. News Articles

Regarding domain identification on news articles, Hamza et al. [20] built a data set containing news articles written in Urdu that were annotated with seven domains as classes according to their topic. Their feature set consisted of unigrams, bigrams, and Term Frequency–Inverse Document Frequency (TF-IDF) values. Following the stages of preprocessing, namely, stopwords removal and stemming, they performed text classification to the seven domains by employing six machine learning models; the Multi-Layered Perceptron (MLP) reached the highest accuracy of 91.4%. Their findings showed that stemming did not positively affect the performance of the models; however, stopwords removal had worsened it. Another paper by Balouchzahi et al. [21] attempted domain identification on fake news articles written in English that were annotated with six domains according to their topic. Their ensemble of RoBERTa, DistilBERT and BERT managed up to 85.5% for the F1 score.

#### 2.1.2. Technical Text

There are certain researchers who performed domain identifications on technical text. Hande et al. [22] classified scientific articles in seven computer science domains by using transfer learning with BERT, RoBERTa, and SciBERT. They found that the ensemble reached its best performance when the weights were taken into account. In the research of Dowla-

gar and Mamidi [23], experiments with BERT and XLM-ROBERTa with a convolutional neural network (CNN) on a multilingual technical data set obtained better results in comparison to experiments with support vector machines (SVM) with TF-IDF and CNN. By selecting the textual data written in Telugu from the same data set, Gundapu and Mamidi [24] obtained up to 69.9% for the F1 score with CNN and a self-attention-based bidirectional long short-term memory (BiLSTM) network.

### 2.1.3. Open Data and Wikipedia Articles

Regarding domain identification on open data, Lalithsena et al. [25] performed automatic topic identification by using MapReduce combined with manual validation by humans on several data sets from Linked Open Data. In order to designate distinct topics, they used specialized tags for the annotation.

In the paper of Nakatani et al. [26], Wikipedia structural feature and term extraction were performed with the aim to score both topic coverage and topic detailedness on web search results that were relevant to the related search queries. Saxena et al. [27] built domain-specific conceptual bases using Wikipedia navigational templates. They employed a knowledge graph and then applied fuzzy logic on each article's network metrics. In the research of Stoica et al. [28], a Wikipedia article by topic extractor was created. Preprocessing included parsing the articles for lower-casing, stopwords removal, and embedding generation. The extractor obtained high precision, recall, and an F1 score of up to 90% with Random Forest, SVM, and Extreme Gradient Boosting (XGBoost), along with cross-validation.

In the authors' previous research, Nikiforos et al. [19], automatic vocational domain identification was performed. A domain-specific textual data set from Wikipedia articles was created, along with a linguistic feature set with TF-IDF values. Preprocessing included tokenization, removal of numbers, punctuation marks, stopwords and duplicates, and lemmatization. Five vocational domains where displaced communities, such as migrants and refugees, commonly seek and find employment were considered as classes. Machine learning experiments were performed with Random Forest combined with AdaBoost and Gradient Boosted Trees, with the latter obtaining the best performance of up to 99.93% accuracy and a 100% F1-score.

In Table 1, the performance of the related work mentioned in this subsection is shown, in terms of evaluation metrics such as accuracy and F1 score, and it considers the data sets and models that procured the best results for each research paper.

**Table 1.** Performance per research paper. Data sets and models of related work with the best results.

| Paper | Classes | Model | Performance |
|---|---|---|---|
| Hamza et al. [20] | 7 domains of Urdu news | MLP | Accuracy: 91.4% |
| Balouchzahi et al. [21] | 6 domains of English fake news | Ensemble: RoBERTa, DistilBERT, BERT | F1 score: 85.5% |
| Hande et al. [22] | 7 computer science domains of scientific articles | Ensemble: BERT, RoBERTa, SciBERT | Accuracy: 92%, F1 score: 98% |
| Dowlagar & Mamidi [23] | 7 multilingual technical domains | BERT, XLM-ROBERTa, CNN | F1 score (macro): 80.3% |
| Gundapu & Mamidi [24] | 6 Telugu technical domains | CNN, BiLSTM | F1 score: 69.9% |
| Stoica et al. [28] | 3 topic domains of Wikipedia | BERT, Random Forest, XGBoost | F1 score: 90% |
| Nikiforos et al. [19] | 5 vocational domains of English Wikipedia | Gradient Boosted Trees | Accuracy: 99.9%, F1 score: 100% |

## 2.2. Social Text Data Scraping

Data scraping and the analysis of textual data from social networks and Wikipedia have been attempted in recent research. "Data analysis is the method of extracting solutions to the problems via interrogation and interpretation of data" [29]. Despite the development of numerous web scrapers and crawlers, social data scraping and analysis of high quality still present a challenging task.

### 2.2.1. Social Networks

Several web scrapers were developed with Python. Scrapy, by Thomas and Mathur [29], scraped textual data from Reddit (https://www.reddit.com/, accessed on 20 October 2022) and stored them in CSV files. Another scraper, by Kumar and Zymbler [30], scraped the Twitter API (https://developer.twitter.com/en, accessed on 20 October 2022) to download tweets regarding particular airlines, which then were used as input for sentiment analysis and machine learning experiments with SVM and CNN, and their results reached up to 92.3% accuracy.

### 2.2.2. Wikipedia

Other web crawlers, more focused on Wikipedia data, were built. iPopulator by Lange et al. [10] used conditional random fields (CRF) and crawled Wikipedia to gather textual data from the first paragraphs of Wikipedia articles and then used them to populate an infobox for each article. iPopulator reached up to 91% in average extraction precision with 1727 infobox attributes. Cleoria and a MapReduce parser were used by Hardik et al. [11] to download and process XML files with the aim to evaluate the linkability factor of Wikipedia pages.

In the authors' previous research, Nikiforos et al. [19], a web crawler was developed using the Python libraries BeautifulSoup4 and Requests. It scraped Wikipedia's API by downloading textual data from 57 articles written in English, wherein it considered as a criterion their relevance to five vocational domains in which refugees and migrants commonly seek and find employment. The aim was to extract linguistic information concerning these domains and perform machine learning experiments for domain identification.

## 2.3. Data Oversampling and Undersampling

Data sampling, either oversampling or undersampling, is one of the proposed solutions to mitigate the class imbalance problem. Resampling techniques practically change the class distribution in imbalanced data sets by creating new examples for the minority class(es) (oversampling), removing examples from the majority class (undersampling), or doing both (hybrid) [12,16].

Several researchers proposed data undersampling techniques. Lin et al. [13] proposed two undersampling strategies in which a clustering technique was applied during preprocessing; the number of clusters of the majority class was made equal to the number of data points of the minority class. In order to represent the majority class, cluster centers and nearest neighbors of the cluster centers were used by the two strategies, respectively. They performed experiments on 44 small-scale and 2 large-scale data sets to result in the second strategy approach, which combined with a single multilayer perceptron and a C4.5 decision tree and performed better compared to five state-of-the-art approaches. Anand et al. [14] introduced an undersampling technique and evaluated it by performing experiments on four real biological imbalanced data sets. Their technique improved the model sensitivity compared to weighted SVMs and other models in the related work for the same data. Yen and Lee [15] proposed cluster-based undersampling approaches to define representative data as the training set with the aim to increase the classification accuracy for the minority class in imbalanced data sets. García and Herrera [16] presented evolutionary undersampling, which is a taxonomy of methods that considers the nature of the problem and then applies different fitness functions to achieve both class balance and high performance. Their experiments with numerous imbalanced data sets showed that evolutionary undersampling performed better than other state-of-the-art undersampling models when the imbalance was increased.

Other researchers experimented with data oversampling and hybrid approaches. Shelke et al. [18] examined class imbalance on text classification tasks with multiple classes, thereby addressing the sparsity and high dimensionality of textual data. After applying a combination of undersampling and oversampling techniques on the data, they performed experiments with multinomial Naïve Bayes, k-Nearest Neighbor, and SVMs. They concluded that the effectiveness of resampling techniques was highly data dependent, while certain resampling techniques achieved better performance when combined with specific classifiers. Lopez et al. [12] provided an extensive overview of class imbalance mitigating methodologies, namely, data sampling, algorithmic modification, and cost-sensitive learning. They discussed the most significant challenges regarding using data intrinsic characteristics, namely, small disjuncts, lack of density in the training set, class overlapping, noisy data identification, borderline instances, and the data set shift between the training and the test distributions in classification problems with imbalanced data sets. Their experiments on imbalanced data led to important observations on the reaction of machine learning algorithms to data with these intrinsic characteristics. One of the most notable approaches is that of Chawla et al. [17]. They proposed a hybrid approach for classification on imbalanced data, which achieved better performance compared to exclusively undersampling the majority class. Their oversampling method, also known as SMOTE, produced synthetic minority class examples. Their experiments were performed with C4.5, Ripper, and Naïve Bayes, while their method was evaluated with the area under the receiver operating characteristic curve (AUC) and the receiver operating characteristic (ROC) convex hull strategy. The SMOTE oversampling method has been used in this paper to balance the data set (Section 5).

## 3. Data Set Creation and Preprocessing

The data set which was used in the authors' conference paper [19] was created by scraping 57 articles written in English from Wikipedia's API (https://pypi.org/project/wikipedia/, accessed on 5 June 2022) with Python (BeautifulSoup4 (https://pypi.org/project/beautifulsoup4/, accessed on 5 June 2022) and Requests (https://pypi.org/project/requests/, accessed on 5 June 2022)). The criterion for selecting these specific articles was their relevance to five vocational domains considered to be the most common for refugee and migrant employment in Europe, Canada, and the United States of America [1,2,6–8].

The initial textual data set comprised of 6827 sentences extracted from the 57 Wikipedia articles. The data set was preprocessed in four stages, namely:

1. Initial preprocessing and tokenization;
2. Numbers and punctuation mark removal;
3. Stopwords removal;
4. Lemmatization and duplicate removal.

The data set was initially tokenized to 6827 sentences and to 69,062 words; the sentences were used as training–testing examples, and the words were used as unigram features. Numbers, punctuation marks, and special characters were removed. Stopwords (conjunctions, articles, adverbs, pronouns, auxiliary verbs, etc.) were also removed. Finally, lemmatization was performed to normalize the data without reducing the semantic information, and 912 duplicate sentences and 58,393 duplicate words were removed. For more details on these stages of preprocessing, refer to [19].

Resulting from the preprocessing stages, the text data set comprised 5915 sentences (examples) and five classes to be used in machine learning experiments. For each sentence, the domain that was most relevant to each article's topic, as shown in Table 2, was considered as its class, thus resulting in five distinct classes, namely: A. Agriculture, B. Cooking, C. Crafting, D. Construction, and E. Hospitality. The distribution of the sentences to these five classes is shown in Figure 1.

A RapidMiner Studio (version 9.10) process, as shown in Figure 2, was used to extract the feature set with TF-IDF values and taking into consideration the feature occurrences by pruning features, which rarely occur (below 1%) or very often occur (above 30%); this resulted

in 109 unigram features. For more details on the operators and parameters of the feature extraction process, refer to [19] and RapidMiner documentation (https://docs.rapidminer.com/, accessed on 20 October 2022). It is important to note that the extracted features that were used as inputs for the machine learning experiments in this paper were terms in the form of single words—also known as unigrams. Unigrams are the most simple and generic linguistic features that can be used in NLP tasks. Consequently, the methodology described in this paper is not overspecified, meaning that it can be generalized and applied in any corpus, and these features can be used as inputs for any machine learning model.

**Table 2.** Wikipedia articles that were scraped to create the data set, shown by domain categorization.

| Agriculture *10 Articles* | Cooking *17 Articles* | Crafting *11 Articles* | Construction *7 Articles* | Hospitality *12 Articles* |
|---|---|---|---|---|
| Agriculture Glossary of agriculture Farm Farmer Environmental impact of agriculture History of agriculture Intensive farming Plant breeding Subsistence agriculture Sustainable agriculture | Al dente Al forno Baking Charcuterie Chef Chef's uniform Chocolate Cooking Cooking school Cooking weights and measures Cuisine Denaturation(food) Garde manger List of cooking techniques Mise en place Outdoor cooking Outline of food preparation | Anvil Blacksmith Bladesmith Coppersmith Forge Goldsmith Gunsmith Locksmithing Metalsmith Silversmith Whitesmith | Building Building design construction Carpentry Construction Constructor worker Glossary of construction costs Home construction | Bellhop Casino hotel Check-in Concierge Doorman(profession) Hostel Hotel Hotel manager Maid Receptionist Resort Tourism |

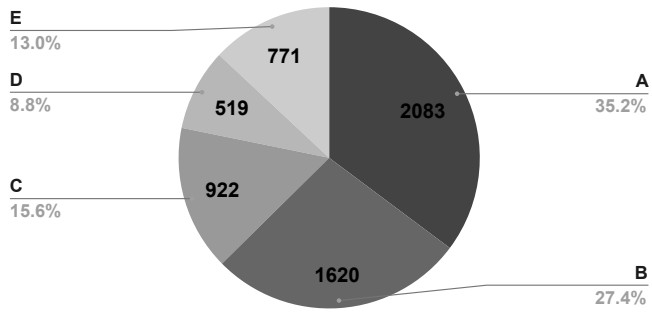

**Distribution of Examples to Classes**

A: Agriculture, B: Cooking, C: Crafting, D: Construction, E: Hospitality

E 13.0% — 771
D 8.8% — 519
C 15.6% — 922
A 35.2% — 2083
B 27.4% — 1620
1620

**Figure 1.** Final distribution of sentences used as training–test examples for the 5 classes.

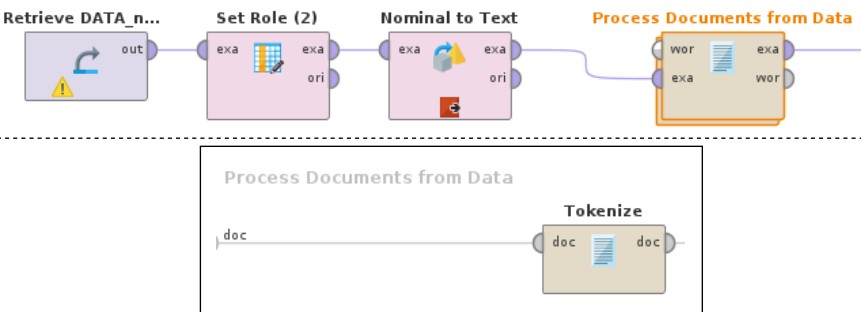

**Figure 2.** Feature extraction. The *Tokenize* operator is nested in the *Process Documents from Data* operator.

Resulting from the feature extraction process, the final data set comprised 5915 examples, 109 features, and a *class* as *label*.

## 4. Predictions Analysis

The best results on domain identification were obtained with a *Gradient Boosted Trees* ([https://docs.rapidminer.com/9.10/studio/operators/modeling/predictive/trees/gradient_boosted_trees.html](https://docs.rapidminer.com/9.10/studio/operators/modeling/predictive/trees/gradient_boosted_trees.html), accessed on 20 October 2022) model and are shown in Table 3 in terms of accuracy ($Accuracy = \frac{TP+TN}{TP+TN+FP+FN}$), precision ($Precision = \frac{TP}{TP+FP}$), recall ($Recall = \frac{TP}{TP+FN}$), and F1 score ($F1 = \frac{2TP}{2TP+FP+FN}$).

**Table 3.** Machine learning experiment results with *Gradient Boosted Trees*. Accuracy: 99.93%.

| Class | Precision | Recall | F1 Score |
|---|---|---|---|
| A | 100% | 100% | 100% |
| B | 100% | 99.94% | 99.97% |
| C | 99.78% | 99.89% | 99.83% |
| D | 99.81% | 99.61% | 99.70% |
| E | 99.87% | 100% | 99.93% |

*Gradient Boosted Trees* is a forward-learning ensemble of either regression or classification models that depends on the task. It uses steadily improved estimations, thus resulting in better predictions in terms of accuracy. More specifically, a sequence of weak prediction models, in this case *Decision Trees*, creates an ensemble that steadily improves its predictions based on the changes in data after each round of classification. This boosting method and the parallel execution running on a H2O 3.30.0.1 cluster, along with the variety of refined parameters for tuning, enable *Gradient Boosted Trees* to be a robust and highly effective model that can overcome issues that are typical for other tree models (e.g., *Decision Trees* and *Random Forest*), such as data imbalance and overfitting. Additionally, it has to be noted that, despite the fact that other methods of tree boosting tend to decrease the speed of the model and human interpretability of its results, the gradient boosting method generalizes the boosting process and, thus, mitigates these problems while maintaining high accuracy.

Regarding the parameters for *Gradient Boosted Trees*, the *number of trees* was set to 50, the *maximal depth* of trees was set to 5, *min rows* was set to 10, *min split improvement* was left at the default, *number of bins* was set to 20, *learning rate* was set to 0.01, *sample rate* was set to 1, and the *distribution* function of the training data was selected automatically as multinomial, since the *label* was nominal for the specific task and data set. For more information on the operators and parameters of the RapidMiner Studio (version 9.10) experiment with *Gradient Boosted Trees*, as shown in Figure 3, refer to [19] and the RapidMiner documentation.

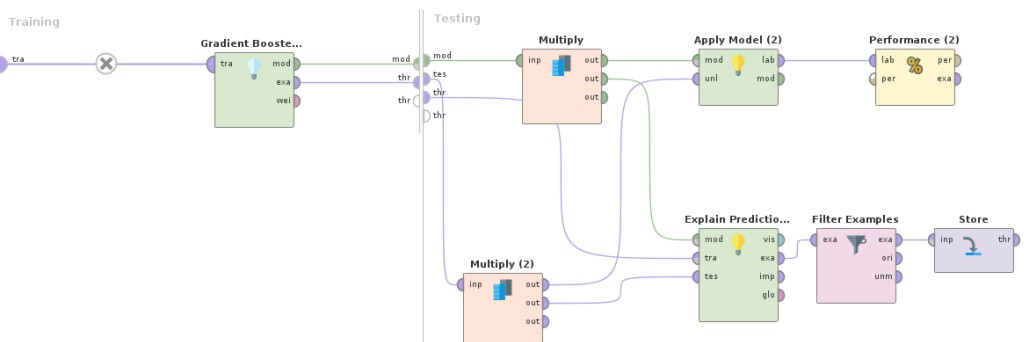

**Figure 3.** Setup of machine learning experiment with *Gradient Boosted Trees*. The depicted process is nested in a *Cross Validation* operator (10-fold cross validation with stratified sampling).

With regard to the high performance of this machine learning model, it is of interest to examine which examples were classified wrongly for each class, as well as which distinct features contributed to their misclassification. In the same line of thought, regarding the correctly classified examples, the examination of the features that were the most dominant and led to correct predictions would contribute to the identification of a primary set of terms that highlighted the terminology of the vocational domains.

### 4.1. Wrong Predictions

The *Gradient Boosted Trees* model showed high performance regarding all classes (Table 3), with a precision ranging from 99.78% to 100%, a recall ranging from 99.61% to 100%, and an F1 score ranging from 99.70% to 100%, and it misclassified a total of four examples. In order to identify the misclassified examples, a RapidMiner Studio (version 9.10) process, as shown in Figure 4, was designed and executed.

**Figure 4.** Setup of process to identify wrong and correct predictions with *Explain Predictions* operator and *Filter Examples* operator. The depicted process is nested in a *Cross Validation* operator (10-fold cross validation).

The *Explain Predictions* (https://docs.rapidminer.com/10.1/studio/operators/scoring/explain_predictions.html, accessed on 20 March 2023) operator was used to identify which features were the most dominant in forming predictions. A model and a set of examples, along with the feature set, were considered as inputs in order to produce a table highlighting the features that most strongly supported or contradicted each prediction, while also containing numeric details. For each example, a neighboring set of data points was generated by using correlation to define the local feature weights in that neighborhood. The operator can calculate model-specific weights though model-agnostic global feature weights that derive directly from the explanations. *Explain Predictions* is able to work with all data types and data sizes and can be applied for both classification and regression problems.

In this case, in which the machine learning model (*Gradient Boosted Trees*) used supervised learning, all supporting local explanations added positively to the weights for correct predictions, while all contradicting local explanations added positively to the weights for wrong predictions. Regarding the parameters for this operator, the *maximal explaining attributes* were set to 3 and the *local sample size* was left at the default (500). The *sort weights* parameter was set to true, along with the descending *sort direction* of the weight values, in order to apply sorting to the resulting feature weights supporting and contradicting the predictions.

The *Filter Examples* (https://docs.rapidminer.com/10.1/studio/operators/blending/examples/filter/filter_examples.html, accessed on 20 March 2023) operator selects which examples are kept and which are removed. In this case, only the misclassified examples (wrong predictions) were kept. Regarding the *condition class* parameter for this operator, it was set to wrong_predictions in order to only keep those examples where the class and prediction were different, which meant that the prediction was wrong.

The four misclassified examples included the following:

1. WP1: building edifice structure roof wall standing permanently house factory;
2. WP2: typically whitesmiths product required decorative finish fire grate coldworking screw lathed machine;
3. WP3: organic food;
4. WP4: traditional vernacular building method suit local condition climate dispensed favour generic cookie cutter housing type.

In Table 4, detailed information is provided for these wrong predictions. Class is the real class of the example, while Prediction is the wrongly predicted class for the example. Confidence, with values ranging from 0 to 1, is derived from feature weights regarding both Class and Prediction.

**Table 4.** Wrong predictions of *Gradient Boosted Trees*. *Class* is the real class of the example, and *Prediction* is the wrongly predicted class for the example. *Confidence*, ranging from 0 to 1, and derived from feature weights regarding both *Class* and *Prediction*, as is shown in the last 2 columns.

| No. | Class | Prediction | Confidence (Class) | Confidence (Prediction) |
|-----|-------|------------|--------------------|-------------------------|
| WP1 | D | C | 0.14 | 0.41 |
| WP2 | C | D | 0.12 | 0.48 |
| WP3 | B | C | 0.11 | 0.55 |
| WP4 | D | E | 0.17 | 0.31 |

The features that contributed to the wrong predictions for each class are shown in Table 5. The effect of the value for each feature was denoted in consideration of whether it supported, contradicted, or was neutral to the prediction. The typical value for the specific feature for each class is also provided.

**Table 5.** Features that contributed to the wrong predictions of *Gradient Boosted Trees*. *Effect* denotes whether the specific value of the specific feature supports, contradicts, or is neutral to the prediction. *Typical Value* is the typical value for the specific feature for each class.

| No. | Feature(s) | Value(s) | Effect(s) | Typical Value |
|-----|------------|----------|-----------|---------------|
| WP1 | building | 1 | Neutral | D: 0 <br> C: 0 and some 1 |
| WP2 | typically <br> fire <br> product | 1 <br> 0.66 <br> 0.54 | Neutral <br> Contradict | C & D: 0 and some 1 |
| WP3 | food <br> organic | 0.50 <br> 0.86 | Support | B: 0 & C: 1 <br> B: 0 & C: 0 and some 1 |
| WP4 | local <br> method <br> type | 0.56 <br> 0.47 <br> 0.46 | Support | D: 0 & E: 0 and some 1 |

*4.2. Correct Predictions*

The *Gradient Boosted Trees* model managed to correctly classify most of the examples. Regarding class A, it is of particular interest that all of its examples were classified correctly,

while none of the examples of the other classes were classified wrongly to class A. Consequently, it is of significance to identify and examine which features were the most dominant and led to the correct predictions for each class, thus contributing to the identification of a primary set of terms for the vocational domains.

In order to identify the correctly classified examples, the same RapidMiner Studio (version 9.10) process, as was used for wrong predictions (Figure 4), was used. The only difference was that the *Condition Class* parameter for the *Filter Examples* operator was set to correct_predictions in order to only keep those examples where the class and prediction were the same, which meant that the prediction was correct.

The Confidence parameter, with values that can be from 0 to 1, was derived from feature weights for each class: for class A, it ranged from 0.49 to 0.55; for class B, it ranged from 0.37 to 0.55; for class C, it ranged from 0.48 to 0.55; for class D, it ranged from 0.47 to 0.55; and, for class E, it ranged from 0.54 to 0.55. The features that were the most dominant and led to the correct predictions are shown in Table 6 in a descending order, along with the global weights that were calculated for each one of them.

**Table 6.** Global weights per feature (descending order). Features with higher weights were more dominant for the correct predictions of this model than features with lower weights.

| No. | Feature | Weight | No. | Feature | Weight |
|-----|---------|--------|-----|---------|--------|
| 1 | farmer | 0.037 | 56 | time | 0.018 |
| 2 | world | 0.036 | 57 | usually | 0.018 |
| 3 | blacksmith | 0.034 | 58 | cocoa | 0.018 |
| 4 | using | 0.034 | 59 | grain | 0.017 |
| 5 | produce | 0.033 | 60 | material | 0.017 |
| 6 | human | 0.032 | 61 | chef | 0.017 |
| 7 | developed | 0.031 | 62 | growing | 0.017 |
| 8 | plant | 0.03 | 63 | process | 0.017 |
| 9 | yield | 0.029 | 64 | water | 0.017 |
| 10 | food | 0.029 | 65 | form | 0.017 |
| 11 | project | 0.029 | 66 | industry | 0.016 |
| 12 | temperature | 0.029 | 67 | fat | 0.016 |
| 13 | environmental | 0.028 | 68 | field | 0.016 |
| 14 | ingredient | 0.028 | 69 | found | 0.016 |
| 15 | america | 0.028 | 70 | domesticated | 0.015 |
| 16 | technique | 0.028 | 71 | product | 0.015 |
| 17 | united | 0.027 | 72 | sometimes | 0.015 |
| 18 | design | 0.027 | 73 | europe | 0.015 |
| 19 | heat | 0.027 | 74 | crop | 0.015 |
| 20 | system | 0.027 | 75 | source | 0.015 |
| 21 | quality | 0.026 | 76 | anvil | 0.014 |
| 22 | iron | 0.026 | 77 | variety | 0.014 |
| 23 | breeding | 0.026 | 78 | various | 0.013 |
| 24 | local | 0.025 | 79 | livestock | 0.013 |
| 25 | vegetable | 0.025 | 80 | tourism | 0.013 |
| 26 | typically | 0.025 | 81 | farm | 0.013 |
| 27 | increase | 0.025 | 82 | construction | 0.013 |
| 28 | land | 0.024 | 83 | practice | 0.013 |
| 29 | cost | 0.024 | 84 | building | 0.013 |
| 30 | agricultural | 0.024 | 85 | people | 0.012 |
| 31 | sustainable | 0.024 | 86 | natural | 0.012 |
| 32 | common | 0.023 | 87 | example | 0.012 |
| 33 | called | 0.023 | 88 | level | 0.012 |
| 34 | service | 0.023 | 89 | animal | 0.012 |
| 35 | period | 0.023 | 90 | organic | 0.012 |
| 36 | cuisine | 0.022 | 91 | soil | 0.011 |
| 37 | trade | 0.022 | 92 | resort | 0.011 |
| 38 | production | 0.022 | 93 | cooking | 0.011 |
| 39 | operation | 0.022 | 94 | meat | 0.011 |

**Table 6.** *Cont.*

| No. | Feature | Weight | No. | Feature | Weight |
|-----|---------|--------|-----|---------|--------|
| 40 | country | 0.022 | 95 | especially | 0.01 |
| 41 | farming | 0.022 | 96 | population | 0.01 |
| 42 | include | 0.021 | 97 | fire | 0.01 |
| 43 | global | 0.021 | 98 | hotel | 0.01 |
| 44 | effect | 0.021 | 99 | modern | 0.009 |
| 45 | increased | 0.021 | 100 | century | 0.009 |
| 46 | type | 0.02 | 101 | change | 0.009 |
| 47 | agriculture | 0.02 | 102 | chocolate | 0.009 |
| 48 | method | 0.02 | 103 | metal | 0.008 |
| 49 | fertilizer | 0.02 | 104 | including | 0.008 |
| 50 | amount | 0.02 | 105 | steel | 0.008 |
| 51 | baking | 0.02 | 106 | smith | 0.007 |
| 52 | tool | 0.019 | 107 | text | 0.007 |
| 53 | oven | 0.019 | 108 | management | 0.006 |
| 54 | worker | 0.018 | 109 | due | 0.006 |
| 55 | hot | 0.018 | | | |

*4.3. Discussion*

Regarding the wrong prediction analysis, the four misclassified examples were successfully identified (WP1–WP4), as shown in Table 4. More specifically, two examples of class D, namely, WP1 and WP4, were wrongly classified to classes C and E, respectively, while one example of class C, WP2, was misclassified to class D, and one example of class B, WP3, was misclassified to class C. It was observed that, for all wrong predictions, the Confidence for the Class, which is the real class of the examples, ranged from 0.11 to 0.17 and was significantly lower than the Confidence for Prediction, which is the wrongly predicted class of the examples and ranged from 0.31 to 0.55. This indicates that these examples diverged significantly from the other examples of their class. By examining Tables 5 and 6, this observation can be explained as described below.

For WP1, the value for the *building* feature was 1, while, typically for examples of D (class), the values were 0 and, of C (prediction), they were mostly 0 and sometimes 1. Considering that *building* was the only most dominant feature of WP1, with an assigned feature weight of 0.013, its overall impact on the prediction being neutral was expected.

For WP2, the value for the *typically* feature was 1, for the *fire* feature was 0.66, and for the *product* feature was 0.54, while, typically, the values of all these features for examples of both C (class) and D (prediction) were mostly 0 and sometimes 1. Considering that *typically* was the most dominant feature of WP2, with an assigned feature weight of 0.025, which is high, its overall impact on the prediction being neutral was expected. The *fire* and *product* features contradicted the prediction, though, due to their quite low feature weights of 0.01 and 0.015, respectively, their effects on the prediction were insignificant.

For WP3, the value for the *food* feature was 0.50 and for the *organic* feature was 0.86, while, typically, for examples of B (class), the values were 0 for both features and, of C (prediction), the value was 1 for the *food* feature and mostly 0 and sometimes 1 for the *organic* feature. Considering that *food* was the most dominant feature of WP3, with an assigned feature weight of 0.029, which is high, its overall impact on the prediction being positive (support) was expected. The *organic* feature also supported the prediction, though, due to its quite low feature weight (0.012), its effect on the prediction was insignificant.

For WP4, the value for the *local* feature was 0.56, for the *method* feature it was 0.47, and for the *type* feature it was 0.46, while, typically, the values of all these features for examples of D (class) were 0 and, for E (prediction), were mostly 0 and sometimes 1. Considering that *local* was the most dominant feature of WP4, with an assigned feature weight of 0.025, which is high, its overall impact on the prediction being positive (support) was expected. The *method* and *type* features also supported the prediction, with quite high feature weights of 0.02 for both, and they had a significant effect on the prediction.

Overall, it became evident that the main factor that led the *Gradient Boosted Trees* model to misclassify the examples was the lack of dominant features supporting the real class more than the prediction in terms of feature weight.

Regarding the correct prediction analysis, it was observed that the confidence for the correct predictions for all classes was considerably high, with the lowest being for class B in a range from 0.37 to 0.55 and the highest for class E in a range from 0.54 to 0.55. This means that the model could classify the examples of class E more confidently compared to the examples of the other classes.

Additionally, the most dominant features, in terms of feature weights, which led to the correct predictions for each class, were identified successfully and sorted in descending order, as shown in Table 6. Features with higher weights were more dominant for the correct predictions of this model than features with lower weights. A total of 51 features, which were about half of the 109 features of the extracted feature set, had the highest feature weights, which ranged from 0.02 up to 0.037. This indicates that the feature extraction process, as described in Section 3 and [19], performed quite well, thus producing a robust feature set with great impact on the correct predictions. Finally, it was also observed that, among these features, terms that were relevant to all of the vocational domains were included, thus yielding a primary set of terms for the vocational domains.

## 5. Data Balancing

Another machine learning experiment on domain identification was performed with a *Random Forest* (https://docs.rapidminer.com/9.10/studio/operators/modeling/predictive/trees/parallel_random_forest.html, accessed on 20 October 2022) and *AdaBoost* (https://docs.rapidminer.com/9.1/studio/operators/modeling/predictive/ensembles/adaboost.html, accessed on 20 October 2022) model. The results of this experiment are shown in Table 7.

**Table 7.** Machine learning experiment results with *Random Forest* and *AdaBoost*. Accuracy: 62.33%.

| Class | Precision | Recall | F1 Score |
|-------|-----------|--------|----------|
| A | 49.06% | 97.60% | 65.29% |
| B | 91.52% | 51.30% | 65.74% |
| C | 95.05% | 41.65% | 57.92% |
| D | 91.67% | 31.79% | 47.20% |
| E | 98.21% | 35.54% | 52.19% |

*Random Forest* is an ensemble of random trees that are created and trained on bootstrapped subsets of the data set. For a random tree, each node constitutes a splitting rule for one particular feature, while a subset of the features, according to a subset ratio criterion (e.g., information gain), is considered for selecting the splitting rules. In classification tasks, the rules are splitting values that belong to different classes. New nodes are created repeatedly until the stopping criteria are met. Then, each random tree provides a prediction for each example by following the tree branches according to the splitting rules and by evaluating the leaf. Class predictions are based on the majority of the examples, and estimations are procured through the average of values reaching a leaf, thus resulting in a voting model of all created random trees. The final prediction of the voting model usually varies less than the single predictions, since all single predictions are considered equally significant and are based on subsets of the data set.

*AdaBoost*, aka Adaptive Boosting, is a meta-algorithm that can be used in combination with various learning algorithms in order to improve their performance. Its adaptiveness is due to the fact that any subsequent classifiers built are adapted in favor of the examples that were misclassified by previous classifiers. *AdaBoost* is sensitive to noisy data and outliers; however, in some tasks, it may be less susceptible to overfitting than most learning algorithms. It is important to note that, even with weak classifiers (e.g in terms of error rate), the final model is improved when its performance is not random. *AdaBoost* generates and calls a new weak classifier in each of a series of rounds $t = 1, \ldots, T$. For each call,

a distribution of weights $D(t)$ is updated. This distribution denotes the significance of examples in the data set for the classification task. During each round, the weights of each misclassified example are increased, while the weights of each correctly classified example are decreased, in order for the new classifier to focus on the misclassified examples.

Regarding the parameters for *Random Forest*, the *number of trees* was set to 100, the *maximal depth* of trees was set to 10, *information gain* was selected as the criterion for feature splitting, and *confidence vote* was selected as the voting strategy. Neither pruning nor prepruning were selected, since it was observed that they did not improve the performance of the model for this task. The maximum *iterations* for *AdaBoost* were set to 10. For more information on the operators and parameters of the RapidMiner Studio (version 9.10) experiment with *Random Forest* and *AdaBoost*, as shown in Figure 5, refer to [19] and the RapidMiner documentation (https://docs.rapidminer.com/, accessed on 20 October 2022).

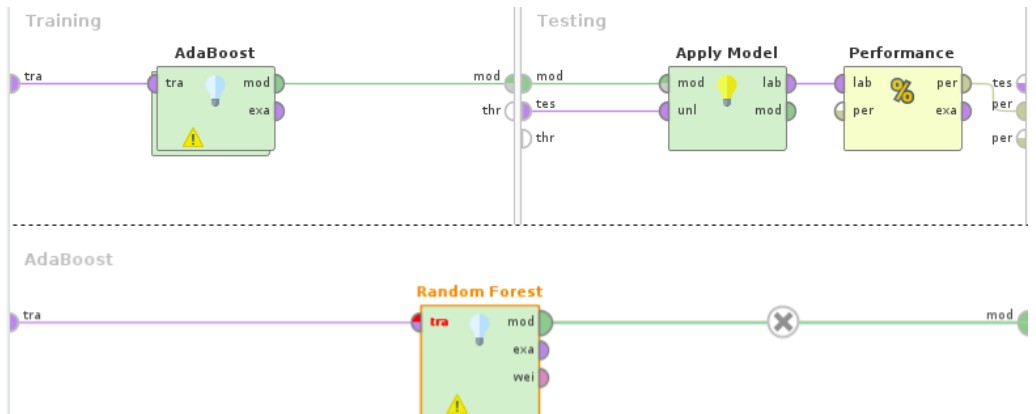

**Figure 5.** Setup of machine learning experiment with *Random Forest* and *AdaBoost*. The *Random Forest* operator is nested in the *AdaBoost* operator. The depicted process is nested in a *Cross Validation* operator (10-fold cross validation with stratified sampling).

Regarding the model's accuracy of 62.33%, it is important to bear in mind that, despite being considerably lower than the accuracy of the *Gradient Boosted Trees* model (99.93%), it was significantly above the randomness baseline by 42.33%, considering that the randomness for a five-class problem was at 20%.

Examining the model's results (Table 7) more closely, it was noted that, despite its precision for classes B, C, D, and E being high, which ranged from 91.52% to 98.21%, the recall for these classes was low, which ranged from 31.79% to 51.30%. Also, considering its low precision (49.06%) and high recall (97.60%) for class A, this examination highlighted that a lot of the examples were classified wrongly to class A. As a result, it became evident that the *Random Forest* and *AdaBoost* model tended to classify most of the examples to class A. Due to the fact that the examples of class A consisted of the majority of the examples in the data set (35.20%, Figure 1), this tendency could be attributed to the imbalance of data.

Consequently, it is of interest to examine whether applying data balancing techniques on the data set (oversampling and undersampling), has any impact, whether positive or negative, on the performance of the *Random Forest* and *AdaBoost* model.

### 5.1. Data Oversampling

As a first step towards addressing data imbalance, SMOTE oversampling [17] was applied in a successive manner on the data set in order to balance the data using oversampling, which pertained to the minority class each time. Consequently, a RapidMiner Studio (version 9.10) process, as shown in Figure 6, was designed and executed four times. The four derived oversampled data sets were then used as inputs for the machine learning experiments with *Random Forest* and *AdaBoost*.

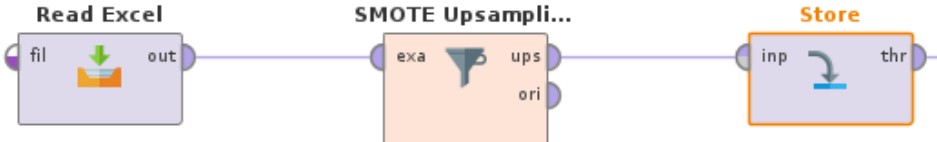

**Figure 6.** Setup of *SMOTE Upsampling*.

The *SMOTE Upsampling* (https://docs.rapidminer.com/10.1/studio/operators/extensions /Operator%20Toolbox/blending/smote.html, accessed on 20 March 2023) operator practically applies the Synthetic Minority Oversampling Technique, as defined in the paper by Chawla et al. [17]. More specifically, the algorithm considers only the examples of the minority class, and the k nearest neighbors for each example are searched. Then, a random example and a random nearest neighbor for this example are selected, thus resulting in the creation of a new example on the line between the two examples.

Regarding the parameters for this operator, the *number of neighbors* was left at the default (5), while *normalize* and *round integers* were set to true, and *nominal change rate* was set to 0.5 in order to make the distance calculation solid. The *equalize classes* parameter was set to true to draw the necessary amount of examples for class balance, and the *auto detect minority class* was set to true to automatically upsample the class with the least amount of occurrences.

The set of machine learning experiments with successive applications of SMOTE oversampling, as described below, follows a novel and original methodology, since it was defined and used for the specific task for the first time, to the best of the authors' knowledge. The methodology steps were the following:

1. Detect the minority class;
2. Resample the minority class with SMOTE oversampling;
3. Run the machine learning experiment;
4. Repeat steps 1–3 until the data set is balanced (no minority class exists).

By running the experiments following this methodology, the impact of every class distribution, from completely imbalanced to completely balanced data, on the performance of the machine learning model could be examined thoroughly. Consequently, this four-step methodology was an important contribution of this paper.

In the first machine learning experiment, class D was the minority class, with its examples representing merely 8.8% of the data set (Figure 1). After applying SMOTE, class D represented 27.9% of the data set with 2083 examples (Figure 7). The results of the *Random Forest* and *AdaBoost* with *SMOTE* are shown in Table 8.

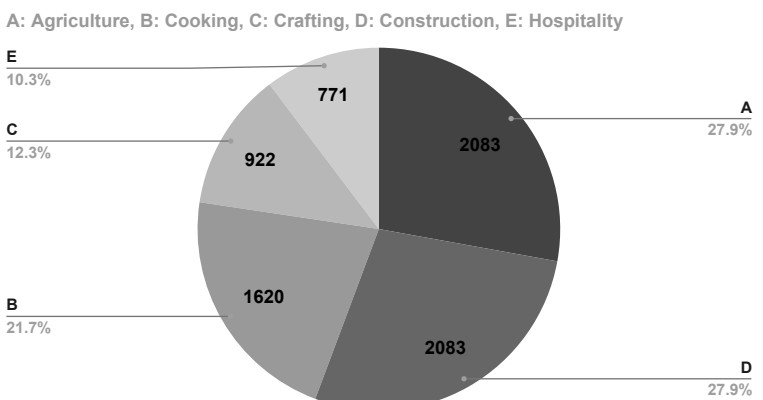

**Figure 7.** Distribution of examples to classes after applying SMOTE (1 time).

**Table 8.** Machine learning experiment results with *Random Forest* and *AdaBoost* with *SMOTE*. Accuracy: 66.01%.

| Class | Precision | Recall | F1 Score |
|---|---|---|---|
| A | 94.30% | 69.13% | 79.77% |
| B | 92.67% | 49.20% | 64.27% |
| C | 94.47% | 38.94% | 55.14% |
| D | 46.65% | 99.33% | 63.48% |
| E | 98.19% | 35.28% | 51.90% |

In the second machine learning experiment, class E was the minority class, with its examples representing 10.3% of the data set (Figure 7). After applying SMOTE, class E represented 23.7% of the data set with 2083 examples (Figure 8). The results of the *Random Forest* and *AdaBoost* with *SMOTE* (two times) are shown in Table 9.

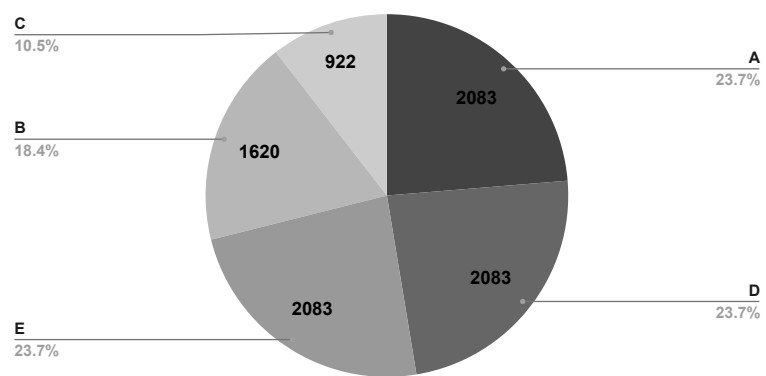

**Distribution of Examples to Classes after SMOTE 2**

A: Agriculture, B: Cooking, C: Crafting, D: Construction, E: Hospitality

**Figure 8.** Distribution of examples to classes after applying SMOTE (2 times).

**Table 9.** Machine learning experiment results with *Random Forest* and *AdaBoost* with *SMOTE* (2 times). Accuracy: 65.72%.

| Class | Precision | Recall | F1 Score |
|---|---|---|---|
| A | 93.46% | 67.93% | 78.67% |
| B | 92.20% | 47.41% | 62.62% |
| C | 94.51% | 37.31% | 53.49% |
| D | 60.50% | 72.35% | 65.89% |
| E | 48.57% | 83.68% | 61.46% |

In the third machine learning experiment, class C was the minority class, with its examples representing 10.5% of the data set (Figure 8). After applying SMOTE, class C represented 20.9% of the data set with 2083 examples (Figure 9). The results of the *Random Forest* and *AdaBoost* with *SMOTE* (three times) are shown in Table 10.

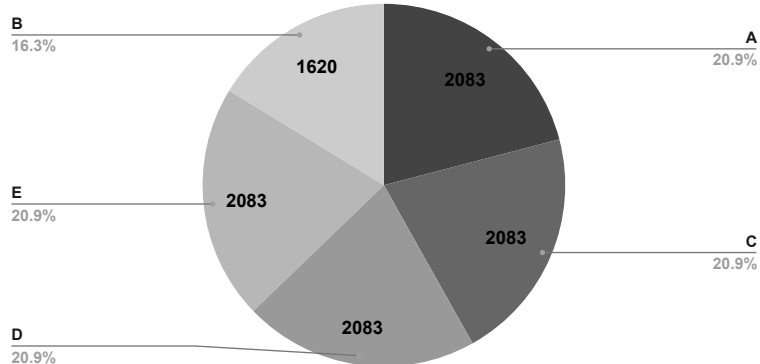

**Figure 9.** Distribution of examples to classes after applying SMOTE (3 times).

In the fourth machine learning experiment, class B was the minority class, with its examples representing 16.3% of the data set (Figure 9). After applying SMOTE, class B represented 20% of the data set with 2083 examples (Figure 10). The results of the *Random Forest* and *AdaBoost* with *SMOTE* (four times) are shown in Table 11.

**Table 10.** Machine learning experiment results with *Random Forest* and *AdaBoost* with *SMOTE* (3 times). Accuracy: 64.35%.

| Class | Precision | Recall | F1 Score |
|-------|-----------|--------|----------|
| A | 95.14% | 66.73% | 78.44% |
| B | 91.72% | 46.48% | 61.69% |
| C | 38.44% | 96.54% | 54.98% |
| D | 89.68% | 58.38% | 70.72% |
| E | 95.48% | 49.64% | 65.32% |

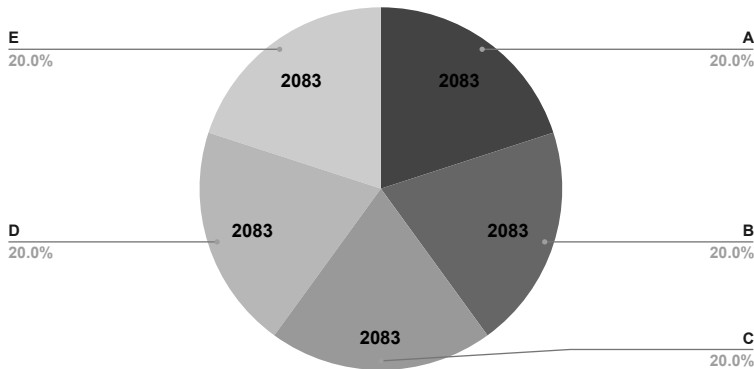

**Figure 10.** Distribution of examples to classes after applying SMOTE (4 times).

**Table 11.** Machine learning experiment results with *Random Forest* and *AdaBoost* with *SMOTE* (4 times). Accuracy: 64.09%.

| Class | Precision | Recall | F1 Score |
|---|---|---|---|
| A | 95.52% | 65.53% | 65.29% |
| B | 40.79% | 85.84% | 65.74% |
| C | 58.84% | 62.17% | 57.92% |
| D | 90.00% | 58.33% | 47.20% |
| E | 96.20% | 48.58% | 52.19% |

*5.2. Data Undersampling*

In another set of experiments, undersampling was applied to the data set, thereby balancing the data by undersampling the classes represented by the most examples. Consequently, a RapidMiner Studio (version 9.10) process, as shown in Figure 11, was designed and executed. The derived undersampled data set was then used as the input for the machine learning experiments with *Random Forest* and *AdaBoost*.

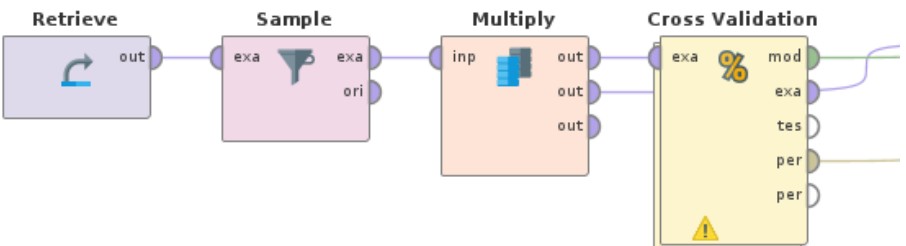

**Figure 11.** Setup of machine learning experiment with *Random Forest* and *AdaBoost* with *Sample*. The *Random Forest* operator is nested in the *AdaBoost* operator, which is nested in a *Cross Validation* operator (10-fold cross validation).

The *Sample* (https://docs.rapidminer.com/10.1/studio/operators/blending/examples/sampling/sample.html, accessed on 20 March 2023) operator has basic principles that are common to the *Filter Examples* operator, wherein it takes a set of examples as the input and procures a subset of it as output. However, while *Filter Examples* follows previously specified conditions, *Sample* is centered on the number of examples and the class distribution in the resulting subset, thus producing samples in a random manner.

Regarding the parameters for this operator, *Sample* was set to absolute in order for it to be created to consist of an exact number of examples. The *Balance Data* parameter was set to true in order to define different sample sizes (by number of examples) for each class, while the class distribution of the sample was set with *Sample Size Per Class*. Examples of classes A and B were reduced to 1183 for each one, which is the mean of the number of all examples in the data set. The sample sizes for each class are shown in Figure 12. The results of this experiment are shown in Table 12.

**Table 12.** Machine learning experiment results with *Random Forest* and *AdaBoost* with *Sample*. Accuracy: 62.84%.

| Class | Precision | Recall | F1 Score |
|---|---|---|---|
| A | 94.34% | 66.27% | 77.85% |
| B | 41.77% | 96.53% | 58.30% |
| C | 94.94% | 44.79% | 60.86% |
| D | 88.57% | 41.81% | 56.80% |
| E | 96.40% | 41.63% | 58.14% |

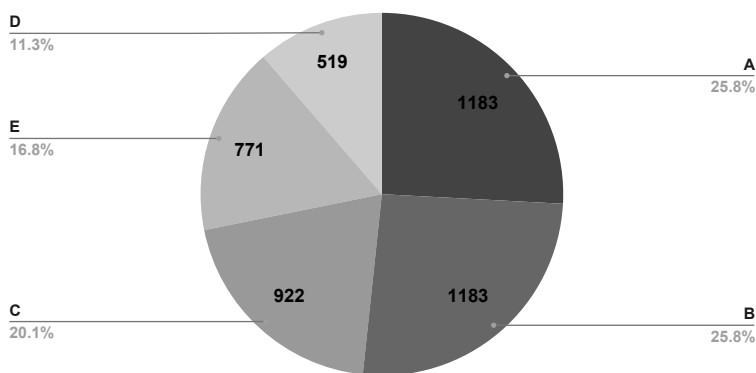

**Figure 12.** Distribution of examples to classes after applying Sample.

A hybrid approach combining data oversampling and undersampling was also tested. In this experiment, both the *SMOTE Upsampling* operator and the *Sample* operator were applied on the data set to balance the data by undersampling the classes represented by the most examples and oversampling the classes represented by the least examples, respectively. Consequently, a RapidMiner Studio (version 9.10) process, as shown in Figure 13, was designed and executed. The derived undersampled data set was then used as the input for the machine learning experiments with *Random Forest* and *AdaBoost*.

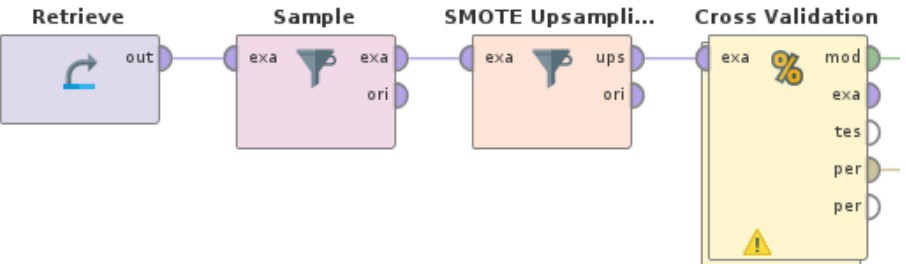

**Figure 13.** Setup of machine learning experiment with *Random Forest* and *AdaBoost* with *Sample* and *SMOTE Upsampling*. The *Random Forest* operator is nested in the *AdaBoost* operator, which is nested in a *Cross Validation* operator (10-fold cross validation).

Regarding the parameters for *Sample* and *SMOTE Upsampling*, they were set in the same way as in the previous experiments. After applying them, examples of classes A and B were reduced to 1183 for each one, which is the mean of the number of all examples in the data set, while examples of class D were added to also result in 1183 for this class. The sample sizes for each class are shown in Figure 14. The results of this experiment are shown in Table 13.

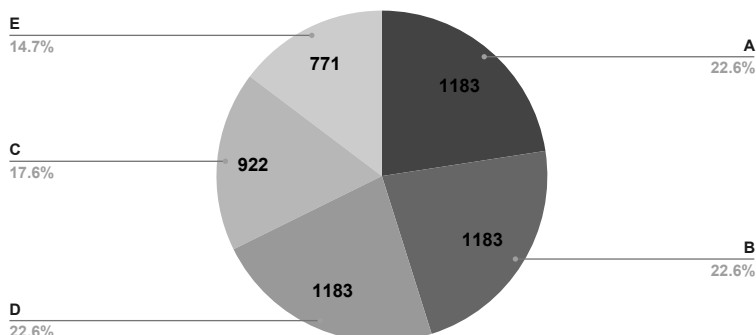

**Figure 14.** Distribution of examples to classes after applying Sample and SMOTE.

**Table 13.** Machine learning experiment results with *Random Forest* and *AdaBoost* with *Sample* and *SMOTE Upsampling*. Accuracy: 63.35%.

| Class | Precision | Recall | F1 Score |
|---|---|---|---|
| A | 95.29% | 66.69% | 78.46% |
| B | 39.79% | 95.52% | 56.17% |
| C | 93.62% | 44.58% | 60.39% |
| D | 83.27% | 57.23% | 67.83% |
| E | 97.52% | 40.73% | 57.46% |

*5.3. Discussion*

Regarding the machine learning experiments' results with *Random Forest* and *AdaBoost* with *SMOTE* oversampling, it was observed that the accuracy and overall performance, as shown in Tables 8–11, improved compared to those of *Random Forest* and *AdaBoost* with imbalanced data, as shown in Table 7. More specifically, the accuracy increased from 62.33% up to 66.01%, and the F1 score increased from 65.29% up to 79.77% for class A, maintained up to 65.74% for class B, maintained up to 57.92% for class C, increased from 47.20% up to 70.72% for class D, and increased from 52.19% up to 65.32% for class E. It is also noteworthy that, despite the overall performance of the model becoming slightly worse with each iteration (each added SMOTE oversampling), it was still significantly better than the performance of the experiment with completely imbalanced data; even the lowest accuracy (64.09%), which was that of the fourth machine learning experiment with SMOTE, was quite higher than the accuracy (62.33%) of the experiment with completely imbalanced data. Additionally, the values of precision, recall, and F1 score seemed to be distributed more evenly among the classes with each iteration, thus mitigating any emerging bias of the model towards one particular class. Another important observation from these experiments is that, in a classification task where one of the five vocational domains may be considered as the class of interest, e.g., for trying to exclusively detect articles of a specific vocational domain from a corpus to filter relevant content, the application of SMOTE oversampling for the class of interest had a positive effect on the results of this classification task.

Regarding the machine learning experiments' results with *Random Forest* and *AdaBoost* with *Sample*, it was observed that the accuracy and overall performance, as shown in Table 12, improved slightly compared to those of *Random Forest* and *AdaBoost* with imbalanced data, as shown in Table 7. More specifically, accuracy increased from 62.33% to 62.84%, and the F1 score increased from 65.29% to 77.85% for class A, reduced from 65.74% to 58.30% for class B, increased from 57.92% to 60.86% for class C, increased from 47.20% to 56.80% for class D, and increased from 52.19% to 58.14% for class E. Compared to the results

obtained with SMOTE oversampling (Tables 8–11), undersampling had worse performance in terms of accuracy, class precision, recall, and F1 score.

Regarding the machine learning experiments' results with *Random Forest* and *AdaBoost* with *Sample* and *SMOTE* oversampling (hybrid approach), it was observed that the accuracy and overall performance, as shown in Table 13, marginally improved compared to those of *Random Forest* and *AdaBoost* with *Sample* only (Table 12). More specifically, the accuracy increased from 62.84% to 63.35%, and the F1 score increased from 77.85% to 78.46% for class A, reduced from 58.30% to 56.17% for class B, reduced from 60.86% to 60.39% for class C, increased from 56.80% to 67.83% for class D, and reduced from 58.14% to 57.46% for class E. In any case, the performance of this experiment was better than that of the experiment with completely imbalanced data. Overall, these experiments indicate that, when applying both data undersampling and oversampling in a hybrid approach, the results were better than only applying undersampling but were worse than only applying oversampling for this data set.

The findings derived from the machine learning experiments of this paper are in accordance with those of the relevant literature [12,17], with the results that data oversampling obtained better results than data undersampling in imbalanced data sets, while hybrid approaches performed reasonably well. The performance of all the machine learning experiments performed in this research is shown in Table 14.

**Table 14.** Performance of all the machine learning experiments performed in this research. GBT: Gradient Boosted Trees, RF: Random Forest.

| Experiment | Accuracy | Precision (Average) | Recall (Average) | F1 Score (Average) |
|---|---|---|---|---|
| GBT | 99.93% | 99.89% | 99.88% | 99.88% |
| RF + AdaBoost | 62.33% | 85.10% | 51.57% | 57.66% |
| RF + AdaBoost – SMOTE 1 | 66.01% | 85.25% | 58.37% | 62.91% |
| RF + AdaBoost – SMOTE 2 | 65.72% | 77.84% | 61.73% | 64.42% |
| RF + AdaBoost – SMOTE 3 | 64.35% | 82.09% | 63.55% | 66.23% |
| RF + AdaBoost – SMOTE 4 | 64.09% | 76.27% | 64.09% | 57.66% |
| RF + AdaBoost – Undersampling | 62.84% | 83.20% | 58.20% | 62.39% |
| RF + AdaBoost – Undersampling + SMOTE | 63.35% | 81.89% | 60.95% | 64.06% |

## 6. Conclusions

Displaced communities, such as migrants and refugees, face multiple challenges in seeking and finding employment in high-skill vocations in their host countries, which derive from discrimination. Unemployment and overworking phenomena usually affect the displaced communities more than the natives. A deciding factor for their prospects of employment is the knowledge of not the language of their host country in general, but specifically of the sublanguage of the vocational domain they are interested in working. Consequently, more and more highly skilled migrants and refugees worldwide are finding employment in low-skill vocations, despite their professional qualifications and educational backgrounds, with the language barrier being one of the most important factors. Both high-skill and low-skill vocations in agriculture, cooking, crafting, construction, and hospitality, among others, consist of the most common vocational domains in which migrants and refugees seek and find employment according to the findings of the recent research.

In the last decade, due to the expansion of the user base of wikis and social networks, user-generated content has increased exponentially, thereby providing a valuable source of data for various tasks and applications in data mining, natural language processing, and machine learning. However, minority class examples are the most difficult to obtain from real data, especially from user-generated content from wikis and social networks, thereby creating a class imbalance problem that affects various aspects of real-world applications that are based on classification. Especially for multi-class problems, such as the one addressed in this paper, they are more challenging to solve.

This paper extends the contribution of the authors' previous research [19] on automatic vocational domain identification by further processing and analyzing the results of machine learning experiments with a domain-specific textual data set, wherein we considered two research directions: a. prediction analysis and b. data balancing.

Regarding the prediction analysis direction, important conclusions were drawn from successfully identifying and examining the four misclassified examples (WP1–WP4) for each class (wrong predictions) using the *Gradient Boosted Trees* model, which managed to correctly classify most of the examples, as well as identify which distinct features contributed to their misclassification. An important finding is that the misclassified examples diverged significantly from the other examples of their class, since, for all wrong predictions, the confidence values for class, which is the real class of the examples, were significantly lower (from 0.11 to 0.17) than the confidence values for prediction (from 0.31 to 0.55), which indicates the wrongly predicted class of the examples. More specifically, the feature values of WP1–WP4 were the main factors for their misclassification, by either being neutral or by supporting the wrong over the correct prediction. Even when they contradicted the wrong prediction, such as the features of WP2 and WP3, they did not have a significant effect due to their feature weights being quite low. In conclusion, the main factor that led the *Gradient Boosted Trees* model to misclassify the examples was the lack of dominant features supporting the real class more than the prediction in terms of feature weight.

In the same line of thought, the examination of the correctly classified examples (correct predictions) resulted in several findings. The confidence values for the correct predictions for all classes were considerably high, with the lowest being from class B (from 0.37 to 0.55) and the highest being from class E (from 0.54 to 0.55), which means that the model could classify the examples of class E more confidently compared to the examples of the other classes. Additionally, the most dominant features, in terms of feature weight, led to the correct predictions for each class being identified successfully and sorted in a descending order; features with higher weights were more dominant for the correct predictions of this model than features with lower weights. Another important finding concerning the most dominant features is the fact that about half of the features of the extracted feature set had the highest feature weights (from 0.02 up to 0.037), therefore indicating that the feature extraction process, as described in Section 3 and [19], performed quite well and produced a robust feature set with great impact on the correct predictions. It is important to note that, among these features, terms relevant to all of the vocational domains were included, thus yielding a primary set of terms for the vocational domains.

Regarding the data balancing direction, oversampling and undersampling techniques, both separately and in combination as a hybrid approach, were applied to the data set in order to observe their impacts (positive or negative) on the performance of the *Random Forest* and *AdaBoost* model. A novel and original four-step methodology was proposed in this paper and used for data balancing for the first time, to the best of the authors' knowledge. It consisted of successive applications of SMOTE oversampling on imbalanced data in order to balance them while considering which class was the minority class in each iteration. By running the experiments while following this methodology, the impact of every class distribution, from completely imbalanced to completely balanced data, on the performance of the machine learning model could be examined thoroughly. This process of data balancing enabled the comparison of the performance of this model with balanced data to the performance of the same model with imbalanced data from the previous research [19].

More specifically, the machine learning experiments' results with *Random Forest* and *AdaBoost* with *SMOTE* oversampling obtained significantly improved overall performance and accuracy values (up to 66.01%) compared to those of *Random Forest* and *AdaBoost* with imbalanced data, all while maintaining or surpassing the achieved F1 scores per class. A major finding is that, despite the overall performance of the model becoming slightly worse with each iteration (each added SMOTE oversampling), it was still significantly better than the performance of the experiment with completely imbalanced data; even the lowest accuracy (64.09%), which was that of the fourth machine learning experiment with SMOTE,

was quite higher than the accuracy (62.33%) of the experiment with completely imbalanced data. Moreover, the values of precision, recall, and F1 score seemed to be distributed more evenly among the classes with each iteration, thus mitigating any emerging bias of the model towards one particular class. Another important finding is that, in a classification task where one of the five vocational domains was considered as the class of interest, e.g., for trying to exclusively detect articles of a specific vocational domain from a corpus to filter relevant content, the application of SMOTE oversampling for the class of interest had a positive effect on the results of this classification task.

The machine learning experiments' results with *Random Forest* and *AdaBoost* with *Sample* showed slightly improved overall performance and accuracy values (62.84%) compared to those of *Random Forest* and *AdaBoost* with imbalanced data, all while surpassing the achieved F1 scores per class, except for from class B. Compared to the results obtained with SMOTE oversampling, undersampling had worse performance in terms of accuracy, class precision, recall, and F1 score. The machine learning experiments' results with *Random Forest* and *AdaBoost* with *Sample* and *SMOTE* oversampling (hybrid approach) showed marginally improved overall performance and accuracy values (63.35%) compared to those of *Random Forest* and *AdaBoost* with *Sample* only, all while surpassing the achieved F1 scores for classes A and D. However, the performance of this experiment was better than that of the experiment with completely imbalanced data. In conclusion, these experiments indicate that, when applying both data undersampling and oversampling in a hybrid approach, the results were better than only applying undersampling but were worse than only applying oversampling for this data set. The findings derived from the machine learning experiments of this paper are in accordance with those of the relevant literature [12,17] regarding the conclusion that data oversampling obtains better results than data undersampling in imbalanced data sets, while hybrid approaches perform reasonably well.

In Table 15, the performance of related work (Section 2 and Table 1) is compared to the performance of this paper in terms of accuracy and F1 score, which considers the data sets and models that obtained the best results for each research. The performance of the *Gradient Boosted Trees* model was quite high when compared to the performance of the models applied in related work. It is important to note that Hamza et al. [20] and Balouchzahi et al. [21] worked with data sets consisting of news articles. Hande et al. [22] used scientific articles, and Dowlagar & Mamidi [23] and Gundapu & Mamidi [24] used sentences from technical reports and papers. As a result, their data sets consist of more structured text compared to the social text of the data set created in this paper, which consists of sentences from Wikipedia. Consequently, the fact that the performance of the models of this paper was the same or higher than the performance of the models of the aforementioned papers is noteworthy. Stoica et al. [28], on the other hand, used Wikipedia articles as the input for their models, while sole sentences were used as the input for the models in this paper. Consequently, the fact that the performance of the models of this paper was higher than their performance is also noteworthy. Regarding *Random Forest*, they combined it with XGBoost and obtained much better results (90% F1 score) compared to the results (79.77% F1 score) of the combination with *AdaBoost* used in this paper, thus indicating that the boosting algorithm is crucial to the performance of the models. Another observation is that the performance of *Random Forest* and *AdaBoost* was improved with SMOTE oversampling compared to the authors' previous research [19]. More specifically, the accuracy increased from 62.33% to 66.01%, and the F1 score increased from 65.74% to 79.77%, thus indicating that oversampling had a positive effect on the performance of the model.

**Table 15.** Performance per research paper. A comparison of related work to this paper in terms of data, models, and best results.

| Paper | Data | Model | Performance |
|---|---|---|---|
| Hamza et al. [20] | News articles in Urdu | MLP | Accuracy: 91.4% |
| Balouchzahi et al. [21] | Fake news articles in English | Ensemble: RoBERTa, DistilBERT, BERT | F1 score: 85.5% |
| Hande et al. [22] | Scientific articles of computer science | Ensemble: BERT, RoBERTa, SciBERT | Accuracy: 92%, F1 score: 98% |
| Dowlagar & Mamidi [23] | Multilingual sentences of technical domains | BERT, XLM-ROBERTa, CNN | F1 score (macro): 80.3% |
| Gundapu & Mamidi [24] | Sentences in Telugu of technical domains | CNN, BiLSTM | F1 score: 69.9% |
| Stoica et al. [28] | Wikipedia articles | BERT, RF, XGBoost | F1 score: 90% |
| Nikiforos et al. [19] | Sentences from Wikipedia articles in English | GBT, RF + AdaBoost | Accuracy: 99.9%, F1 score: 100%, Accuracy: 62.33%, F1 score: 65.74% |
| This paper | Sentences from Wikipedia articles in English | GBT, RF + AdaBoost – SMOTE 1 | **Accuracy: 99.9%, F1 score: 100%, Accuracy: 66.01%, F1 score: 79.77%** |

Potential directions for future work include the automatic extraction of domain-specific terminology to be used as a component of an educational tool for sublanguage learning regarding specific vocational domains in host countries with the aim to help displaced communities, such as migrants and refugees, overcome language barriers. This terminology extraction task could use the terms (features) that were identified in this paper as the most dominant for vocational domain identification in terms of feature weight. Moreover, a more vocational domain-specific data set could be created to perform a more specialized domain identification task in vocational subdomains, especially considering the set of dominant terms identified in this paper. Another direction for future work could be performing experiments with a larger data set, wherein they consist of either more Wikipedia articles or even textual data from other wikis and social networks as data sources, in order to examine the impact of more data on the performance of the models. Using a different feature sets, e.g., with n-grams and term collocations, or using features that are more social-text-specific could also be attempted to improve performance. Additionally, machine learning experiments with more intricate boosting algorithms and sophisticated machine learning models could be performed. Finally, another potential direction could be the application of the novel methodology of successive SMOTE oversampling proposed in this paper in combination with undersampling techniques on other imbalanced data sets in order to test its performance in different class imbalance problems.

**Author Contributions:** Conceptualization, K.L.K. and A.P.; methodology, M.N.N.; experiments, M.N.N.; validation, M.N.N.; formal analysis, M.N.N.; investigation, M.N.N.; resources, K.D.; data curation, K.D.; writing—original draft preparation, M.N.N.; writing—review and editing, K.L.K. and A.P.; visualization, M.N.N.; supervision, K.L.K. and A.P. All authors have read and agreed to the published version of the manuscript.

**Funding:** This research received no external funding.

**Institutional Review Board Statement:** Not applicable.

**Informed Consent Statement:** Not applicable.

**Data Availability Statement:** The data are available in a publicly accessible repository that does not issue DOIs. Publicly available data sets were analyzed in this study. This data can be found

here: https://hilab.di.ionio.gr/wp-content/uploads/2023/04/Wikipedia-Articles-for-Vocational-Domain-Identification.xlsx (accessed on 20 May 2023).

**Conflicts of Interest:** The authors declare no conflict of interest.

## Abbreviations

The following abbreviations are used in this manuscript:

| | |
|---|---|
| AdaBoost | Adaptive Boosting |
| AUC | Area Under the Receiver Operating Characteristic curve |
| BiLSTM | Bidirectional Long Short-Term Memory |
| CNN | Convolutional Neural Network |
| CRF | Conditional Random Field |
| Extreme Gradient Boosting | XGBoost |
| FN | False Negative |
| FP | False Positive |
| GBT | Gradient Boosted Trees |
| MLP | Multi-Layered Perceptron |
| NLP | Natural Language Processing |
| RF | Random Forest |
| ROC | Receiver Operating Characteristic curve |
| SVM | Support Vector Machine |
| TF-IDF | Term Frequency - Inverse Document Frequency |
| TN | True Negative |
| TP | True Positive |
| WP | Wrong Prediction |

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
