# Peer review of "Vocational Domain Identification with Machine Learning and Natural Language Processing on Wikipedia Text: Error Analysis and Class Balancing"

_computers, doi:10.3390/computers12060111_

Round 1

Reviewer 1 Report

The paper presents several machine learning experiments performed in RapidMiner Software that used Gradient Boosted Trees, Random Forest and AdaBoost models for classification of texts into vocational domains. These domains correspond to sublanguages in certain professional areas and the existing of such sublanguage may be are barrier for a newcomer such as a migrant or refugee to get a job. The paper is well written and is easy to follow. The figures are clear and helpful, the experiments were well designed and their results support the conclusions drawn. The paper may be accepted.

Reviewer 2 Report

This paper discusses automatic vocational domain identification by analyzing the results of the machine learning experiments with the domain-specific textual data set. The paper claims two contributions. Firstly, the gradient-boosted trees model is studied, where the misclassification results and feature importance are studied. Secondly, various data balancing techniques, such as oversampling and undersampling, are applied to the data set in order to observe their impact on the performance of the Random Forest and AdaBoost models.
There are a few major concerns regarding the contributions of this paper.
1. This is a case study-style paper where existing machine learning models are used in the study. There is no new machine learning model proposed in this paper. The studies are too specific to be applicable to the selected machine learning models (e.g., random forest, adaboost), so the findings cannot be applied to other models.
2. The data balancing techniques are well known in this area. It is not clear what specific challenges are exclusive to be applied to the selected machine learning model in this study.
3. There is no further action proposed. For example, some features are found to have higher impacts on the model prediction. What is the next step to improving the model? This is not discussed.

na

Reviewer 3 Report

Although it is mentioned that different machine learning tools were used, at no time are the algorithms included, or the equations that could give a more formal version in the description of the methodology used, as well as the results obtained.

Neither are the equations of the metrics used to validate the results included, and it is necessary to understand which version of the metrics are being reported in this manuscript.

Finally, a comparison with results of the methods implemented in similar investigations in the state of the art is not presented.

A revision of the English is needed in some sections of the article, because there are some phrases or expressions that are incomplete.

Round 2

Reviewer 2 Report

NA

The revision is fine, there is no further comments.

Reviewer 3 Report

Most of the comments were attended by the authors, therefore, my decision is to accept the manuscript, after a minor revision of the English language.

Most of the comments were attended by the authors, therefore, my decision is to accept the manuscript, after a minor revision of the English language.